# SETD7 Expression Is Associated with Breast Cancer Survival Outcomes for Specific Molecular Subtypes: A Systematic Analysis of Publicly Available Datasets

**DOI:** 10.3390/cancers14246029

**Published:** 2022-12-07

**Authors:** Fátima Liliana Monteiro, Lina Stepanauskaite, Cecilia Williams, Luisa A. Helguero

**Affiliations:** 1Department of Medical Sciences, Institute of Biomedicine—iBiMED, University of Aveiro, 3810-193 Aveiro, Portugal; 2SciLifeLab, Department of Protein Science, KTH Royal Institute of Technology, 114 28 Stockholm, Sweden; 3Department of Biosciences and Nutrition, Karolinska Institute, 141 83 Stockholm, Sweden

**Keywords:** SETD7, breast cancer, molecular subtypes, survival, gene expression, biological processes

## Abstract

**Simple Summary:**

Breast cancer is the most common cancer among women, and it can be classified into subtypes with distinct biology and prognosis. The aim of our bioinformatic study was to assess the potential role of the protein methyltransferase SETD7 in breast cancer by using freely available resources. We saw that SETD7 is differentially expressed across subtypes, which may determine how SETD7 modulates cancer cell biological processes in each subtype. This translates into different prognosis and therapeutic response in patients stratified according to SETD7 levels. SETD7 might provide valuable additional information for discriminating patients based on subtypes and improve therapeutic decisions.

**Abstract:**

SETD7 is a lysine N-methyltransferase that targets many proteins important in breast cancer (BC). However, its role and clinical significance remain unclear. Here, we used online tools and multiple public datasets to explore the predictive potential of *SETD7* expression (high or low quartile) considering BC subtype, grade, stage, and therapy. We also investigated overrepresented biological processes associated with its expression using TCGA-BRCA data. *SETD7* expression was highest in the Her2 (*ERBB2*)-enriched molecular subtype and lowest in the basal-like subtype. For the basal-like subtype specifically, higher *SETD7* was consistently correlated with worse recurrence-free survival (*p* < 0.009). High SETD7-expressing tumours further exhibited a higher rate of *ERBB2* mutation (20% vs. 5%) along with a poorer response to anti-Her2 therapy. Overall, high SETD7-expressing tumours showed higher stromal and lower immune scores. This was specifically related to higher counts of cancer-associated fibroblasts and endothelial cells, but lower B and T cell signatures, especially in the luminal A subtype. Genes significantly associated with SETD7 expression were accordingly overrepresented in immune response processes, with distinct subtype characteristics. We conclude that the prognostic value of SETD7 depends on the BC subtype and that SETD7 may be further explored as a potential treatment-predictive marker for immune checkpoint inhibitors.

## 1. Introduction

SETD7 is a lysine N-methyltransferase that monomethylates the histone H3 lysine 4 (K4) and several other nonhistone proteins, including numerous transcription factors and epigenetic regulators (reviewed in [1]). Methylation by SETD7 can modulate a protein’s stability, subcellular localization, and/or interactions with other proteins. For example, methylation by SETD7 improves the stability of ERα (*ESR1*) in breast cancer (BC), which may be of relevance to endocrine resistance [2]. SETD7 may also be important to prevent oxidative stress in BC cells by reducing KEAP1 and enhancing the expression of *GSTT2* and *NFE2L2* (Nrf2), and promote metastasis by enhancing *VEGFA* or *RUNX2* expression [3]. On the other hand, SETD7 can methylate oncogenic proteins (DNMT1, E2F1, and HIF1A), leading to their degradation (reviewed in [4]). Data obtained mainly from preclinical models point toward a context-dependent effect mediated by SETD7, and its role in BC remains controversial [4].

Several studies have compared SETD7 mRNA or protein expression between BC and nontumorous tissue using public datasets or in-house cohorts, but with inconclusive results. Some studies showed that *SETD7* mRNA levels are lower in BC [5,6], others that BC has higher SETD7 protein levels [7,8] or that no differences in *SETD7* expression between BC and normal tissue were observed [9]. The discrepancy between the studies also translates to the correlation analysis of SETD7 expression with prognosis. While several studies found that high *SETD7* mRNA levels correlated with better overall survival (OS) [10] or disease-free survival (DFS) [11], others reported that higher SETD7 levels were correlated with shorter OS and DFS [3,7,8]. Breast tumours are classified into distinct molecular subtypes based on gene expression (PAM50) profiles. The subtypes encompass luminal A, luminal B, Her2 (*ERBB2*)-enriched, normal-like, and basal-like subtypes, and exhibit fundamental differences in response to therapy and survival. Notably, in all studies that have analysed public datasets, no clarification as to whether the analysis was done by pooling all BC subtypes was available. Upon considering the number of cases in the studies analysing TCGA data, it appears that all subtypes were pooled, but this remains to be clarified in the other studies.

In a recent systematic review, we were able to associate high SETD7 activity with inhibition of epithelial–mesenchymal transition in all the cancer types where this process had been studied, including BC [4]. Moreover, inhibition of SETD7 function was associated with improved response to DNA-damaging agents in most of the analysed studies [4]. Thus, while effects mediated by SETD7 are cell type- and signalling context-specific, the lack of clarity regarding the role and clinical significance of SETD7 in BC may lead to stagnation in this field of research before clear conclusions can be drawn. Herein, an unbiased systematic analysis of public datasets was carried out. The main goal was to establish the predictive potential of *SETD7* expression in BC considering the impact of clinical factors such as subtype, grade, stage, and therapy on the association between *SETD7* expression and survival outcomes. The mutation frequency of *SETD7* and its target proteins was also investigated. Additionally, we identified the significantly overrepresented biological processes and pathways among the differentially expressed genes that emerge when *SETD7* expression is used to stratify the samples in each breast cancer subtype.

## 2. Materials and Methods

### 2.1. Description of Datasets

This study used previously published and publicly available data. No new sequencing or protein expression data were generated. A description of all the datasets that report *SETD7* expression, available for analysis within the different online tools used, is provided in Appendix A. Since cBioPortal (https://www.cbioportal.org/, v5.1.10, accessed on 5 August 2022) includes TCGA breast cancer data with different release dates, we used the PanCancer Atlas study for all analyses in cBioPortal (v5.1.10).

### 2.2. Analysis of SETD7 Mutation and Copy Number

*SETD7* mutation and copy number were analysed using cBioPortal [12,13]. All datasets including mutation and copy number profiles (Appendix A) were pooled and analysis was carried out pooling samples from all BC subtypes [14,15,16,17,18,19,20,21,22,23,24,25,26], including data from The Metastatic Breast Cancer Project (https://www.mbcproject.org/, accessed on 5 August 2022) Count Me In (https://joincountmein.org/, accessed on 5 August 2022) (MBCproject cBioPortal data version February 2020).

### 2.3. Analysis of SETD7 Expression Using Online Tools

The analysis of *SETD7* expression in tumour and adjacent normal tissue was performed using RNA-seq data available in the TNMplot [27] online tool (https://www.tnmplot.com, accessed on 26 April 2022). The relationship between *SETD7* mRNA expression and clinicopathological characteristics, genomic alterations, DNA methylation, phosphoproteome, acetylproteome, and total proteome was explored using cBioPortal [12,13] online tool accessed between 20 January and 27 February 2022. RNA-seq and gene-chip data in cBioPortal are based on z-scores relative to all samples precomputed from the expression values in each dataset (fragments per kilobase of exon per million mapped fragments (FPKM), transcripts per million (TPM), or RNA-seq by expectation maximization (RSEM) for RNA-seq and log(microarray) for gene-chip). *SETD7* differential expression was set by comparing upper vs. lower quartiles (high and low expression, respectively). This analysis was done both by pooling all BC subtypes and for each subtype individually.

### 2.4. Correlation of SETD7 with Breast Cancer Outcomes

KM plotter, cBioPortal, and the Human Protein Atlas (HPA, [28]) were used to study the prognosis value of *SETD7* mRNA or corresponding protein. ROC plotter (https://www.rocplot.org/, accessed on 23 March 2022) [29] was used to study the potential predictive value of SETD7, using the recommended JetSet method [30] and without the ‘no outliers’ filter. ROC plotter uses 36 publicly available BC datasets that include chemotherapy (n = 2108), endocrine therapy (n = 971), and anti-Her2 (n = 267) treatment data. The patients are grouped into responders or non-responders by taking into consideration either the pathological complete response (n = 1775, incl. 639 responders and 1136 non-responders) or the relapse-free survival (n = 1329, incl. 978 responders and 351 non-responders) data provided by the studies. Differential expression of *SETD7* was set by comparing upper (high expression) vs. lower (low expression) quartiles, with exception of protein data in KM plotter and mRNA data in HPA where the differential expression was automatically set (median). Outcomes (OS, RFS—recurrence/relapse-free survival, PCR—pathological complete response, PFS—progression-free survival, DFS, DSFS—disease-specific free survival, DMFS—distant metastasis-free survival, PPS—palliative performance scale) could be evaluated in specific datasets depending on the patient data available for each dataset. Samples grouped by clinical factors or pooled BC subtypes were analysed.

### 2.5. Genes Associated with Differential SETD7 mRNA Expression in BC Subtypes

The TCGA-BRCA raw counts and FPKM data were downloaded on 20 March 2022 from NCI Genomic Data Commons (GDC) using the TCGAbiolinks package (version 2.22.4) [31,32,33] in R (version 4.1.2). SETD7 was defined as highly or lowly expressed based on upper and lower quartiles, respectively. The samples corresponding to the middle quartiles were considered unchanged and therefore removed. Genes with less than 1 FPKM in both high- and low-SETD7 patients were considered not expressed and removed. Genes that were not present in at least a quarter of the samples were also filtered out. This was done based on counts per million using edgeR package (version 3.36.0) [34,35,36]. The raw counts for the remaining samples and genes were then processed using the default processing pipeline of DESeq2 (package version 1.34.0) [37]. Genes were considered significantly expressed if the Benjamini–Hochberg adjusted *p*-value (or *q*-value) for false-discovery rate (FDR) <0.05 and the absolute value of the log2 fold change >0.4. Principal component analysis (PCA) on gene expression (after variance stabilizing transformation to the count data) was used for data visualization. The infiltrating immune and stromal scores for samples expressing high- and low-SETD7 groups were calculated using immunodeconv (version 2.0.4) package [38]. Significantly differentially expressed genes between the high and low SETD7 groups from each BC subtype were extracted using Venny 2.1 [39]. Gene ontology enrichment analysis was performed using DAVID [40,41]. Genes associated with high- or low-SETD7 groups were analysed regardless of direction (up or down) and also separately in an attempt to distinguish which functional results are a subject of *SETD7* expression. The default parameters with medium stringency were used. Biological processes containing at least two annotations and with adjusted *p*-value ≤ 0.05 are reported. The ggplot2 (version 3.3.5) [42] and GOplot (version 1.0.2) [43] packages were used for visualization.

## 3. Results

### 3.1. Characterization of SETD7 Mutations, Copy Number, and Expression in BC

#### 3.1.1. SETD7 Mutation and Copy Number Profile

The frequency of *SETD7* mutations in BC was explored in publicly available data consisting of 8177 samples from 14 independent studies (whole exome sequencing, targeted sequencing, gene chip) [14,15,16,17,18,19,20,21,22,23,24,25,26]. *SETD7* was mutated in only 0.2% of BC cases (7/4378 profiled samples, Appendix A). These rare events corresponded to missense mutations of unknown significance, found randomly across the *SETD7* gene and across subtypes (Appendix A). *SETD7* copy number was altered with a slightly higher frequency of 12% (972/8177 patients). Shallow deletion (heterogeneous loss) of *SETD7* was observed in 17% of cases (761/4378 profiled samples, Appendix A), whereas deep deletion (deep loss, possibly a homozygous deletion) was only observed in 0.1% (3/4378), low-level gains (a few additional copies, often broad) in 4% (188/4378) and high-level amplification (more copies, often local) in 0.5% (23/4378) of cases (Appendix A). The shallow deletion was more often associated with the basal-like subtype (around 47% for basal-like vs. 35% for Her2 -enriched, 21% for luminal A, 33% for luminal B, and 4% for normal-like subtypes). The low-level gain was more often associated with the Her2-enriched subtype (around 13% vs. 5% for basal, 6% for luminal A, 9% for luminal B, and 3% for normal-like subtypes). Overall, the genetic alteration of *SETD7* was not a common occurrence in BC, but a heterogeneous copy number loss was frequent (47%) in basal-like tumours specifically.

#### 3.1.2. Association of SETD7 Expression with Clinical Attributes

To compare the expression of *SETD7* in breast tumours and adjacent normal tissue, we used the publicly available online tool TNMplot comprising RNA-seq data of paired tissue samples from 112 patients. This analysis clearly showed that *SETD7* mRNA is significantly lower in breast tumours compared with the adjacent normal tissue (Figure 1A). Analysis by subtype was not supported by this tool. Next, the expression of *SETD7* was explored in the different BC datasets available from cBioPortal (RNA-seq, gene chip, and mass spectrometry). A significant correlation could be observed between *SETD7* expression and PAM50 subtype (in all datasets except METABRIC [*q* = 0.07], Table 1). *SETD7* mRNA and protein expression were both consistently higher in the Her2-enriched and luminal A subtypes, and lower in the basal subtype (Figure 1B,C and Appendix A, Table 1). The mean differences of each group (Cohen’s d) and the confidence interval for TCGA-BRCA data grouped by subtype were further analysed (Appendix A). Luminal A vs. Her2-enriched (d = −0.25 [−0.54,0.04]), luminal B vs. Her2-enriched (d = −0.07 [−0.39, 0.25]), and luminal B vs. luminal A (d = 0.18 [−0.02, 0.38]) differences had a small effect size indicating little or no clinical relevance. However, the differences between normal-like and luminal B (d = −0.49 [−0.92, −0.06]), Her2-enriched (d = −0.56 [−1.03, −0.08]) or basal subtype (d = 0.36 [−0.07, 0.79]) had a medium effect size, and most importantly, luminal A vs. basal (d = 0.67 [0.46, 0.88]) or luminal B vs. basal (d = 0.85 [0.60, 1.10]), and Her2-enriched vs. basal (d = 0.92 [0.60, 1.25]) had strong effect sizes, supportive of relevant clinical differences (Appendix A). Luminal B tumours exhibited varying mRNA levels dependent on the dataset (Appendix A, Table 1) and low protein levels (Figure 1C). No correlation between *SETD7* differential expression and therapy, tumour grade, or stage was observed in pooled BC samples or when divided by subtype (Appendix A). In conclusion, our analysis across different large-scale datasets clearly shows that *SETD7* expression is significantly reduced in basal-like BC, which may be related to the copy number loss noted above for basal-like tumours.

#### 3.1.3. Association of SETD7 Expression with Clinically Relevant Signatures

To investigate whether *SETD7* differential expression was correlated with clinically relevant mRNA and protein signatures, we first analysed datasets in cBioPortal where this information was available (hypoxia scores were available for the TCGA PanCancer cohort, and stromal, immune, and stemness scores for CPTAC cohort), and as a second approach, we used a deconvolution method in R to further explore the tumour microenvironment infiltration mRNA signatures in the TCGA-BRCA cohort. High *SETD7* mRNA correlated with lower hypoxia scores [44,45] in pooled samples from all subtypes in the TCGA PanCancer Atlas dataset (Figure 2A, left panel). High SETD7 protein correlated with higher stromal scores [46,47] in CPTAC dataset and high *SETD7* mRNA with cancer-associated fibroblasts (CAFs), endothelial cells, and neutrophil signatures in the TCGA-BRCA dataset (Figure 2B). On the other hand, low *SETD7* mRNA and protein levels were correlated with high xCell immune score and stemness score (CPTAC, pooled samples from all subtypes; Table 2 and Figure 2A, middle and right panels). Further analysis of the xCell immune score showed enrichment of B and T cells (CD8+ T cells) in the low-SETD7 group, while, as mentioned above, enrichment of neutrophils was noted in the high-SETD7 group (TCGA-BRCA; Figure 2B).

Analysis by molecular subtype revealed that the lower hypoxia scores in high *SETD7* mRNA group was specific for luminal A and B subtypes (Table 2). Likewise, when each subtype was investigated separately, high stromal scores were significantly correlated with high *SETD7* expression in the luminal A subtype (Table 2) whereas enrichment of CAF, endothelial cell, and neutrophil signatures was noted in high-SETD7 samples of all subtypes (Appendix A).

In conclusion, we saw that reduced levels of SETD7 are associated with high stemness and immune scores in general, while high expression is associated with increased stromal score, including for CAFs, endothelial cells, and neutrophils, and reduced hypoxia scores in the luminal A subtype specifically.

### 3.2. Association of SETD7 Expression with Genomic Alterations and DNA Methylation

SETD7 histone methyltransferase can influence chromatin remodelling. The association of differential *SETD7* expression with genomic alterations and DNA methylation was explored in cBioPortal, using the TCGA PanCancer Atlas cohort to investigate associations with genomic alterations and METABRIC to analyse the impact on DNA methylation. No gene was significantly deleted or mutated in either high- or low-SETD7 mRNA groups, even when specific mutation types were queried individually (missense, in-frame, truncating, structural variants, or CNA deletion), although there was a tendency for *TP53* (p53) gene alterations. Significant correlations between *SETD7* differential expression and other genomic alterations (such as amplifications) were observed (Appendix A). The high-SETD7 group showed higher genomic alterations in *ERBB2* (Her2; 21% event frequency in the high-SETD7 group vs. 6% in the low SETD7 group; Appendix A), especially higher copy number amplification (16.42% or 44/268 profiled samples in high SETD7 compared with 4.85% or 13/268 profiled samples in low SETD7 group). On the other hand, the low-SETD7 group showed higher event frequency in the *TP53* gene (c.a. 28% in high SETD7 vs. 48% in low SETD7, respectively; Appendix A).

Analysis by molecular subtype did not disclose any significant correlation between *SETD7* differential expression and genomic alterations, although a clear tendency for a higher number of genomic alterations in the *ERBB2* gene in the high-SETD7 group was observed for the Her2-enriched subtype (90% event frequency in high SETD7 vs. 58% in low SETD7. Most of these alterations were copy number amplification (27/29 profiled samples).

SETD7 impacts cancer-related processes, including in BC [4,49], but it was not mutated in BC (as shown above). Therefore, mutations in SETD7 target genes were queried. Forty-two specific genes with known SETD7 target methylation sites were analysed (reviewed in [1,4] and detailed in Appendix A). Only one mutation in a SETD7 lysine methylation site was found, consisting of a K873E missense mutation in the tumour suppressor *RB1*, in only one sample. No mutations on sites previously reported to compete with SETD7 methylation [4] were identified.

The correlation with general DNA methylation of SETD7 target genes was investigated in the METABRIC dataset, which is the only set with information about DNA methylation. No correlation between *SETD7* differential expression and DNA methylation throughout the genome was observed, either when pooling all BC samples or when stratifying by molecular subtype.

Thus, high *SETD7* expression was related to increased *ERBB2* copy number in the Her2-enriched subtype, but not related with other genetic alterations.

### 3.3. Gene Expression and Biological Processes Associated with Differential SETD7 mRNA Expression

To avoid heterogeneity, this analysis was carried out on the TCGA data, which is the most powerful gene expression dataset available to date. We retrieved the TCGA-BRCA RNA-seq data, and analysis of differential gene expression between the high-SETD7 and low-SETD7 groups was carried out for each molecular subtype (Appendix A). The normal-like subtype was not included in the analysis due to the low number of samples available. First, the overall gene expression data were validated by the PCA plot clearly separating the basal-like and the Her2-enriched, luminal A and B subtypes, and further showing that the luminal A and B were more similar than the other subtypes, as expected (Appendix A).

Next, the comparison between high- and low-SETD7 groups for each molecular subtype (Venn diagram in Appendix A) disclosed 2834 genes that were commonly associated with *SETD7* expression in all subtypes (Appendix A). Of these, 1699 were highly expressed in the high-SETD7 group and 1133 in the low-SETD7 group. Only two genes (GPER1 and CYP4F22) were oppositely correlated with *SETD7* expression in different subtypes, being enriched in in low-SETD7 tumours for all subtypes except luminal B where they were upregulated in the high-SETD7 group. The commonly upregulated genes in high-SETD7 groups of all subtypes were overrepresented for biological functions related to protein phosphorylation and ubiquitination. Processes previously associated with SETD7 (reviewed in [4]) also appeared overrepresented in the high-SETD7 groups. These include cellular response to DNA damage stimulus, DNA repair, cell division, cell cycle, and cell migration (Appendix A). The genes upregulated in low-SETD7 tumours, on the other hand, were related to translation and mitochondrial respiration (Appendix A).

Further, the unique genes being differentially expressed between high- or low-SETD7 groups within each subtype were analysed for enrichment of biological pathways (Figure 3 and Appendix A). In luminal A subtype, the pathways related with immune response were overrepresented in low-SETD7 group. The highly expressed genes that related more strongly (|log2FC| > 1) with low-SETD7 were mainly immunoglobulins, such as *IGKV2-29* (*q*-value = 7.39^−05^, log2FC = −1.57) and other genes which trigger the immune response such as *AZU1* (*q*-value = 2.24^−16^, log2FC = −1.81) and *S100A9* (*q*-value = 2.72^−11^, log2FC = −1.63) (Appendix A). On the other hand, pathways overrepresented in high-SETD7 tumours were linked to cell adhesion-related pathways (Appendix A). These included the genes *FGB* (*q*-value = 5.99^−03^, log2FC = 1.37) and *ROBO2* (*q*-value = 2.54^−06^, log2FC = 1.32). In the luminal B subtype, DNA repair and response to DNA damage-related pathways were the main biological processes overrepresented in low-SETD7, while the lipid catabolic process was strongly overrepresented in high-SETD7 tumours. Interestingly, magnesium ion transmembrane transport and regulation of insulin secretion involved in cellular response to glucose stimulus were solely overrepresented in the luminal B subtype, where the genes *PNPLA3* (*q*-value = 5.46^−03^, log2FC = 1.27) and *ADCY5* (*q*-value = 4.23^−03^, log2FC = 1.18) stand out as strongly correlated with SETD7 mRNA expression. In the Her2-enriched subtype, fibroblast migration and extracellular matrix disassembly were overrepresented in low-SETD7 tumours, from which the genes *MMP7* (*q*-value = 3.25^−02^, log2FC = −1.32), *KLK5* (*q*-value = 6.67^−03^, log2FC = −1.85), and *KLK7* (*q*-value = 8.63^−03^, log2FC = −2.31) were strongly regulated (|log2FC| > 1). On the other hand, early endosome to late endosome transport was overrepresented in the high-SETD7 group. Finally, in the basal-like subtype, cell differentiation-related pathways were strongly overrepresented in low-SETD7 (Appendix A) where keratins stand out (e.g., *KRT13*, *q*-value = 5.42^−13^, log2FC = −4.36; *KRT6A*, *q*-value = 3.21^−10^, log2FC = −3.56; and *KRT1, q*-value = 8.19^−08^, log2FC = −2.93), along with other genes, such as *SNAI2* (*q*-value = 4.82^−04^, log2FC = −1.13) and *IGF2* (*q*-value = 1.25^−02^, log2FC = −1.01). In the high-SETD7 basal-like tumours, the cellular response to DNA damage stimulus and DNA repair-related pathways were overrepresented. Also, *PPARGC1A* (*q*-value = 1.49^−03^, log2FC = 1.48), a protein involved in cancer metabolic adaptation to stress, was upregulated.

Some biological processes were shared between subtypes, even though the genes for each subtype were unique (Figure 4A). These included many processes related to immune responses, such as chemotaxis, neutrophil chemotaxis and chemokine-mediated signalling, B cell receptor signalling pathway, and inflammatory response. The genes associated with immune response processes were often associated with low-SETD7 in luminal A and Her2-enriched subtypes and with high-SETD7 in the basal-like subtype (Figure 4B and Appendix A). This aligns with the xCell immune score, B and T cell signatures shown above (Figure 2A,B).

Next, the expression of genes with reported functional connection to SETD7 was investigated. A list of 83 genes, including the 42 known SETD7 targets plus other genes reported to be associated with SETD7 function, was used to query the genes differentially expressed in high- versus low-SETD7 tumours for each subtype (Figure 5). Interestingly, some of these genes were consistently associated with high-SETD7 (*AR*, *CTNNB1*, *CTNND1*, *EGFR*, *FOXO3*, *HIF1A*, *HK2*, *KAT2B*, *MED1*, *NFE2L2*, *PDPK1*, *PPP1R12A*, *RB1*, *RORA*, *SIRT1*, *SPEN*, *STAT3*, *YAP1*, and *ZEB1*) or low-SETD7 (*E2F1*, *IRF1*, *MMP7*, *MMP9*, *RPL29*, *SUV39H1*, *TAF10*, *TWIST1*, and *ZFHHC8*) independently of the subtype. Others were dependent on the subtype: *CCNA1*, *DNMT1*, *PPARGC1A*, and *TTK* were associated with high-SETD7 and *SNAI2* with low-SETD7 for the basal-like subtype; *LDHA* and *TP53* with high-SETD7 and *ESR1* (ERα) and *SOX2* with low-SETD7 for the Her2-enriched subtype. Moreover, in both luminal A and B subtypes, *ESR1* (ERα) and *PGR* (PR) were associated with high-SETD7, highlighting an association of SETD7 with endocrine treatment-predictive biomarkers.

In summary, our analysis showed that *SETD7* differential expression is correlated with the expression of different genes depending on the subtype, which may correspond to completely different biological processes (like cell adhesion for luminal A and lysosome organization and early endosome to late endosome transport for Her2-enriched subtype) or shared processes (like immune-related pathways). While no analysis per subtype showed significant results using CPTAC proteome data, it is to be noted that this is a relatively small cohort (total 122 samples), which does not reach a high power when dividing the samples by subtype. Thus, expanding this cohort would be beneficial for further studies.

### 3.4. Association between SETD7 Expression and Its Target Proteins

SETD7 is a methyltransferase with multiple known target proteins. Thus, the protein levels, phosphorylation, and acetylation patterns of SETD7 targets were investigated in *SETD7*-high and -low groups, respectively. The proteins enriched in high- or low-SETD7 groups (all BC subtypes pooled, mRNA data from TCGA PanCancer Atlas and protein data from CPTAC) were extracted and compared with a list of 42 known SETD7 targets (Appendix A). Nineteen targets were not found or were not significantly associated with *SETD7* expression in any dataset (light grey shade in Appendix A). For low-SETD7 tumours, PARP-1 was present in all datasets from CPTAC; Cullin 1 in the total proteome; centromere protein C, HIV Tat (*HTATSF1*), RIO1, DNMT1, and TTK in total and phosphoproteome; Msx2-interacting protein (*SPEN*), catenin beta-1 (*CTNNB1*), TAF7, PPP1R12A, and SUV39H1 in phosphoproteome; MED1 and YY1 in phosphoproteome and acetylproteome; and Sam68 (*KHDRBS1*) and STAT3 in acetylproteome. For high-SETD7 tumours, PPP1R12A, GLI3, YAP1, and AR were found in total and phosphoproteome from CPTAC; and pRb (*RB1*), RELA, ERα, MECP2, and SPEN in phosphoproteome. AR and ERα were also found in the TCGA PanCancer Atlas data.

When analysing the samples per subtype, only the total proteome showed significant correlations with *SETD7* mRNA or protein levels, mainly in the luminal A subtype (Appendix A), where ERα was associated with high *SETD7* mRNA expression (Appendix A). No correlations between *SETD7* differential protein levels and phosphoproteome or acetylproteome were observed by subtype.

### 3.5. Association of SETD7 Expression Levels with Breast Cancer Survival Outcomes

The prognostic value of SETD7 in pooled samples from all BC subtypes was explored using the KMplotter online tool, HPA, and the datasets containing survival data available from cBioPortal. The association of high- or low-SETD7 groups with RFS, DMFS, and OS was variable, varied between datasets, and did not show a clear association with either good or bad prognosis (Appendix A). This is in line with our recent findings reported in a systematic review [4]. Notably, a cohort analysing only ER (*ESR1*)-negative tumours showed that high-SETD7 was significantly correlated with a poor prognosis. This led us to analyse the influence of clinical factors, including the molecular subtype, on the outcome of patients divided according to high or low *SETD7* expression.

#### 3.5.1. Influence of Histological and Molecular Subtype on Outcomes Associated with SETD7 Expression

Survival outcomes were available in METABRIC and TCGA PanCancer Atlas datasets available at cBioPortal. Luminal A patients from the TCGA PanCancer Atlas cohort (244/499 total samples; RNA-seq data) exhibited a correlation between high *SETD7* mRNA and worse OS (*p* = 0.044) and PFS (*p* = 0.032) (Appendix A). However, when all microarray studies were combined (gene-chip data in KM plotter) high-SETD7 correlated with good DMFS. In the luminal B subtype, high-SETD7 correlated with bad DMFS only in one independent study (Appendix A). In the basal-like subtype, significant associations between high *SETD7* expression and worse RFS (gene-chip data; all the studies pooled, Figure 6A, left panel; Appendix A), DMFS and OS (one individual study). Analysis based on high SETD7 protein data [50] in KM plotter also showed that expression was associated with worse OS for ER-negative samples (33/65 total samples, *p* = 0.008; Figure 6A, right panel).

In conclusion, strong evidence suggests that expression of *SETD7* is predictive of a poor outcome for patients carrying basal-like tumours, even though this subtype has lower *SETD7* expression in comparison to the luminal A or Her2-enriched subtypes (Figure 1B,C). For all other subtypes, an association between SETD7 expression and survival outcomes remains inconclusive.

#### 3.5.2. Influence of SETD7 Expression on Therapy Outcomes

Pooled gene-chip studies with information about therapy in KM plotter showed that for patients that only received chemotherapy, high *SETD7* mRNA was significantly correlated with bad RFS (106/211 total samples; *p* = 0.0006; Figure 6B, left panel) and DMFS (84/168 total samples; *p* = 0.0012). The same was observed when analysing two of the studies independently (Appendix A). Most of the patients who received chemotherapy only had basal-like (~60%) or Her2+ (~40%) tumours and, interestingly, high *SETD7* was correlated with worse RFS (Figure 6B, right panels) or DMFS (not shown) only for patients with the basal-like subtype. In patients receiving solely endocrine therapy, high *SETD7* was correlated with worse RFS but in only one study (METABRIC: 495/1025 total samples, *p* = 0.0325; Appendix A). For patients that had not received any therapy or that had received both endocrine and chemotherapy, the results were inconclusive (Appendix A). Thus, SETD7 could be a marker of chemoresistance for patients with basal-like tumours.

#### 3.5.3. Influence of Tumour Stage on Outcomes Associated with SETD7 Expression

TCGA data analysed through the HPA showed that high *SETD7* expression (mean expression) was correlated with low survival for stage II patients (609 samples; *p* = 0.0003). The same was observed using cBioPortal, where higher *SETD7* expression was correlated with lower OS (313/628 total samples; *p* = 0.0283; Figure 6C, left panel) and DFS (312/628 total samples; *p* = 0.0569; Appendix A) for stage 2 in TCGA PanCancer Atlas (confirming the results obtained in the same dataset using HPA; Appendix A) and lower RFS (400/979 total samples; *p* = 0.0396) for stage 2 in METABRIC (Figure 6C, right panel). High SETD7 protein expression [50] in KM plotter was also correlated with low OS (*p* = 0.036) for stage 2 patients (46/65 total samples; Appendix A). It is important to note that stage 2 represents ~60% of all BC samples analysed, followed by stage 1 (~30%) and 3 (~10%). Stages 0 and 4 comprise the lowest percentages of tumours in the cohorts studied and analyses on these were limited. Also, most of the stage 2 tumours in these studies were of the luminal subtype (~60%). Even though one might assume these results suggest that SETD7 could serve as a prognostic marker for luminal tumours in stage 2, this was not confirmed when we pooled stage 2 samples of luminal subtypes from TCGA PanCancer or METABRIC cohorts (not shown).

#### 3.5.4. Influence of Tumour Grade, Lymph Node Status, and Metastasis on Survival Outcomes Associated with SETD7 Expression

The influence of BC grade or lymph node status on the correlation of *SETD7* expression with survival outcomes was not clear, since few independent studies allowed this analysis, and the results did not agree (Appendix A). No association between *SETD7* expression (lower vs. upper quartile) and metastasizing tumours was observed using MBC project in cBioPortal (not shown).

#### 3.5.5. Predictive Power of SETD7

Using ROC plotter, no strong association between *SETD7* differential expression and hormone or chemotherapies was observed. However, patients that did not respond to anti-Her2 therapy expressed higher levels of *SETD7* (Figure 7). This may correlate with the higher *ERBB2* mutation rate in patients of the high-SETD7 group.

## 4. Discussion

Current knowledge on SETD7’s impact on BC biology and its prognostic and predictive potential is scarce, with numerous contradictory findings [4,51]. In this work, we systematically analysed public datasets of BC samples to establish if *SETD7* expression is correlated with, or indicative of, diverse clinical conditions. The relevant biological processes associated with expression of *SETD7*, the genes involved, and their clinical significance were also evaluated. Stratification by molecular subtype, which has not previously been performed, showed that *SETD7* expression was dependent on subtype and that distinct processes were related to *SETD7* expression and could be clinically relevant.

Previous studies comparing *SETD7* expression between normal breast tissue and BC have not reported consistent results. We found that *SETD7* is significantly lower in BC than in adjacent normal tissue (TNMplot). This agrees with previous studies analysing mRNA [5,6] but not with studies analysing the protein level [7,8]. We observed a divergent relationship between mRNA and protein levels specifically for the luminal B subtype. This suggests that SETD7 may be regulated post-transcriptionally, possibly by miR-372/373 [52], or post-translationally, possibly through TRIM21 [8]. For the remaining subtypes, *SETD7* mRNA and protein followed the same pattern. We observed a significantly higher expression in the Her2-enriched and luminal A compared with the basal-like subtype, which may be clinically relevant. This differential *SETD7* expression may be related to the higher frequency of *SETD7* gene loss that we noted for the basal-like subtype and, to some extent, the low-level gain of *SETD7* that we observed among the Her2-enriched subtype tumours.

A relationship between SETD7 expression and prognosis was consistent only for patients with basal-like tumours, where high-SETD7 was significantly associated with worse RFS, DMFS, and OS. This aligned with worse OS for ER-negative patients expressing high SETD7 protein and with worse OS and DMFS in basal-like patients treated with chemotherapy. Higher *SETD7* mRNA and protein in Her2-enriched tumours was correlated with increased *ERBB2* amplification and corresponding *ERBB2* mRNA upregulation. This was associated with a significantly lower response to anti-Her-2 therapy in this subgroup. Still, no significant association with disease prognosis was found, but a trend of poorer OS and RFS could be observed in TCGA Pan Cancer Atlas dataset. Regarding luminal tumours, high SETD7 was also correlated with worse RFS, but only in patients receiving endocrine therapy and this association was not sufficiently strong, as the ROC plot did not support a prognostic value.

As the differential expression of *SETD7* between molecular subtypes may be clinically meaningful, we compared the transcriptomes between high- and low-SETD7 groups. This showed how SETD7 differential expression could impact the biology of the different molecular subtypes and reinforced the experimental data showing that SETD7 function is context-dependent [4]. Thus, the hypothesis raised by this study should be validated in clinical specimens and stratified by molecular subtype. Although many genes associated with SETD7 expression were different depending on subtype, in some cases, the same biological processes were overrepresented. This includes many immune-related processes. While immunotherapy has been increasingly used to treat cancer patients, this line of treatment has not been effective in BC, although some success has been noted for the triple-negative breast cancer subtype (mostly comprising the basal-like subtype) [53,54]. Herein, we show that the immune infiltration and response were highly correlated with *SETD7* expression, especially in luminal A and basal-like subtypes. The corresponding genes were primarily upregulated in low-SETD7 luminal A tumours and also correlated with higher xCell Immune score (represented by signatures of B and T cells). Additionally, the upregulation of genes with a functional role in immune evasion (*PD1*, *FOXP3*, *CTLA4*, *IL17B* and the *IL17* receptors *IL17RE* and *IL17RC*) in the low-*SETD7* group of luminal A subtype supports the knowledge that lymphocyte infiltration is associated with worse prognosis in luminal subtypes [55,56]. Immunotherapy is not currently viewed as relevant in luminal A tumours; however, stratification by SETD7 might improve the response rate of immune checkpoint inhibitors. On the contrary, in the basal-like subtype, immune-related genes were upregulated in the high-SETD7 group, and this was correlated with higher T cell infiltration, including of CD8+, as inferred from their gene expression signatures. This usually corresponds to a better prognosis in the basal-like subtype [55,56], and thus suggests that the tumours expressing high SETD7 might benefit from immunotherapy. However, future studies are needed to verify if stratification by SETD7 alone or together with additional markers can improve selection of patients for immunotherapy in subgroups of luminal A and basal-like tumours.

In luminal subtypes, the two gold-standard biomarkers *ESR1* (ERα) and *PGR* (PR) were associated with high SETD7. ERα is the target of endocrine treatments and a primary treatment-predictive marker in breast cancer [57]. However, many ERα-positive tumours develop endocrine resistance, where ERα is active in the absence of ligand. ERα is a known target of SETD7 [2], which stabilizes ERα through methylation in lysine 302. It is not known if this stabilization contributes to endocrine resistance. However, the lower survival of luminal A high-SETD7 patients from the TCGA dataset, along with the upregulation of *RUNX2* and *GPER1* (in luminal B, with reported roles in breast carcinogenesis [58] and endocrine resistance [59]), suggest a role for SETD7 in endocrine resistance. The idea of targeting SETD7 to overcome endocrine resistance thus deserves further testing. This may be specifically relevant for the luminal B subtype, where the most significant biological processes overrepresented in the high-SETD7 (mRNA) group were the ubiquitin-dependent ERAD pathway and the positive regulation of autophagy, which is also linked to the ubiquitin–proteosome system (UPS, also overrepresented). The association with high SETD7 was not strong, but given that these two pathways underly endocrine resistance [60,61], the additive contribution of all these genes to these processes should not be discarded. Autophagy is associated with the suppression of tumour initiation [62] and the survival of dormant BC stem cells and metastatic tumour recurrence [62,63]. Many preclinical studies have shown that autophagy inhibition improves endocrine therapy response [64]. Although the ROC plotter did not find a correlation between high *SETD7* mRNA stratification and 5-year RFS (suggestive of resistance to endocrine therapy), we need to consider that we could not show a correlation between *SETD7* mRNA and protein levels in this subtype. Further, these results deserve further validation, as the majority of patients were treated with tamoxifen.

In the Her2-enriched subtype, lysosome organization and early endosome to late endosome transport were overrepresented in the high-*SETD7* group. Activation of these two biological processes has been linked to anti-Her2 therapy resistance [65,66]. Remarkably, *EGFR* and *ERBB3* were strongly associated with high SETD7 and also connected to anti-Her2 therapy resistance [66]. These results together with the correlation of high SETD7 with higher amplification of *ERBB2* corroborates the ROC plotter results, where high SETD7 was correlated with patients that did not respond well (shorter 5-year RFS) to anti-Her2 therapy. A recent study suggests SORL1 to be a candidate therapeutic target to complement and potentiate anti-Her2 therapy [67]. Indeed, *SORL1* expression was significantly associated with high SETD7 expression, pointing to a potential benefit of targeting SETD7 alone or together with SORL1 in patients with high SETD7 in order to overcome resistance.

A limitation of our study is that multivariable adjustment was unavailable in the tools used to analyse the survival and prognosis value of SETD7. Moreover, information to correct for confounding effects was lacking, which restricts the conclusions that can be drawn about *SETD7* expression as an independent factor of diagnosis or resistance to therapies. Further studies will be needed to validate the clinical impacts.

In previous preclinical studies, SETD7 function has consistently been associated with altered cellular response to DNA damage stimulus, including hypoxia and oxidative stress and independently of *TP53* status [4]. Genes related to the cellular response to DNA damage stimulus and DNA repair were common to all subtypes (Appendix A). Two of the major players in the DNA damage response pathway, *ATR* and *ATM* were highly expressed in the high-SETD7 group in all subtypes. This indicates that chemotherapy might be less efficient in tumours expressing high SETD7. This was also supported by the poor prognosis associated with SETD7 expression when analysing all BC subtypes from TCGA. When analysis was carried out by BC subtype, the basal-like subtype also showed unique genes associated with *SETD7* expression and strongly overrepresented in cellular response to DNA damage and DNA repair-related pathways were. This was in line with poor outcome after chemotherapy for patients with basal-like tumours expressing high SETD7. In the future and based on previous findings showing that inhibition of SETD7 in other types of cancer improves response to chemotherapy [10,68,69,70,71,72], it would be interesting to explore if this subgroup of patients could benefit from targeting SETD7 with inhibitors to improve chemotherapy response.

## 5. Conclusions

*SETD7* expression appears strongly associated with tumour stromal and immune signatures and related to therapy resistance. In the basal-like BC subtype, high *SETD7* expression was consistently predictive of bad prognosis, and this group was enriched in immune signatures. The unique genes associated with *SETD7* expression were strongly overrepresented in cellular response to DNA damage and DNA repair-related pathways, and this was aligned with poor outcome after chemotherapy. Future studies should focus on the identification of the differentially expressed genes that could constitute markers to aid decisions on prescribing immune therapy and test if inhibiting SETD7 improves basal-like response to chemotherapy. In the Her2-enriched subtype, high SETD7 may also have a predictive value, since SETD7 expression was associated with *ERBB2* copy number amplification and worse response to anti-Her2 therapy, as well as upregulation of EGFR, HER-3, and overrepresentation of such biological processes as lysosome organization and early endosome to late endosome transport known to be underlying mechanisms of anti-Her-2 therapy resistance. In the luminal subtype, high SETD7 expression was associated with higher *ESR1* (ERα), *PGR* (PR), *RUNX2* and *GPER1*, which together with previous findings on the role of SETD7 maintaining ERα protein stability and activity, highlight the need for further studies on the role of SETD7 in endocrine resistance. Still, no consistent relationship with prognosis was found, except for worse OS in tumours with high SETD7 and treated with endocrine therapy.

In summary, this study emphasizes that there is clinical potential in the study of SETD7, which must be evaluated in the context of the BC molecular subtype.

## Figures and Tables

**Figure 1 cancers-14-06029-f001:**
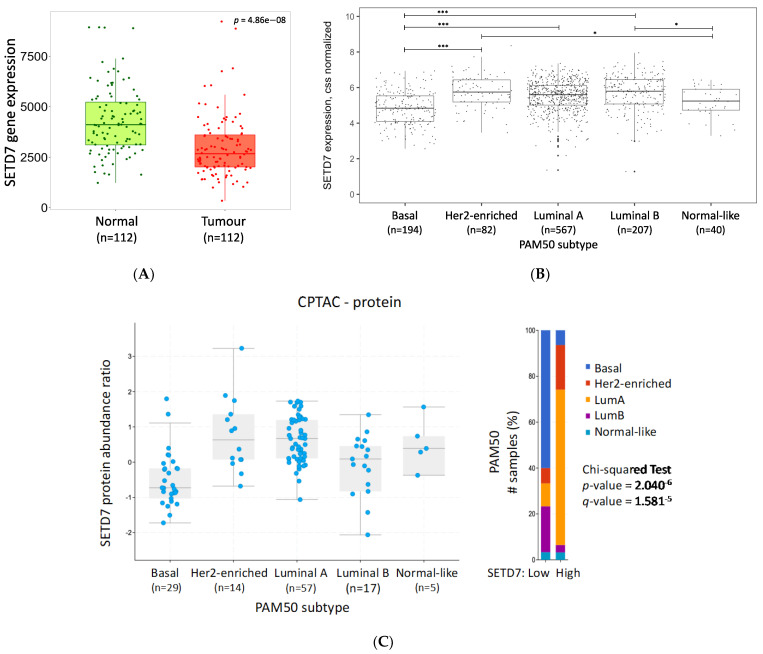
SETD7 expression in breast cancer. (**A**) *SETD7* mRNA expression in tumour and adjacent normal tissue using RNA-seq data available from TNMplot; (**B**) *SETD7* mRNA expression across PAM50 subtypes using TCGA-BRCA data in R. ANOVA followed by Tukey’s test: * < 0.05; *** < 0.0001 (**C**) SETD7 protein expression across PAM50 subtypes using CPTAC data from cBioPortal.

**Figure 2 cancers-14-06029-f002:**
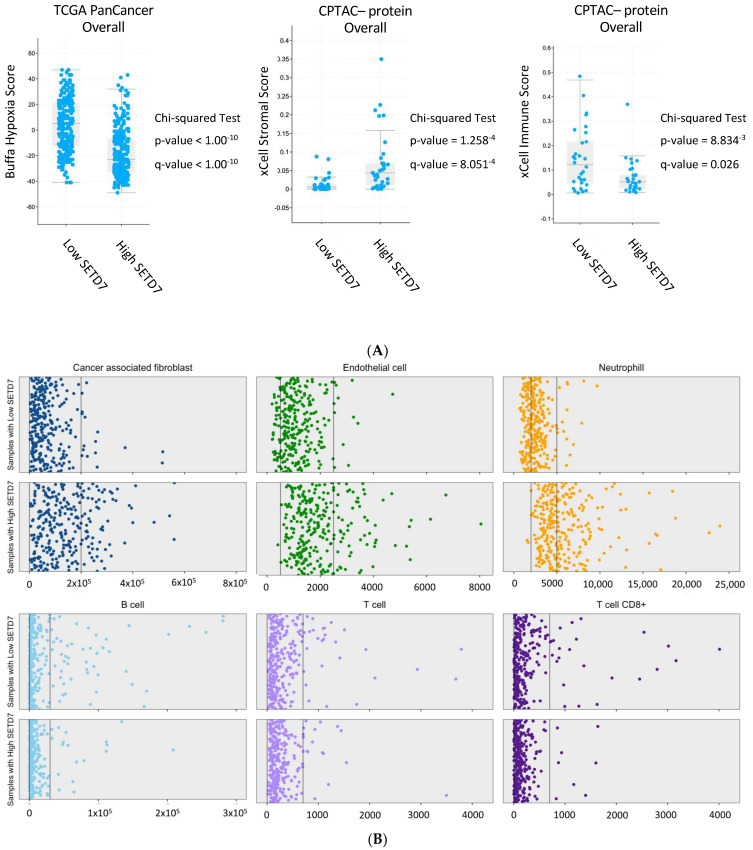
Correlation of *SETD7* differential expression with clinical factors. (**A**) Association of *SETD7* differential expression (high vs. low) with Buffa Hypoxia Score (mRNA), xCell Stromal Score and xCell Immune Score (protein) when pooling all breast cancer types together. Wilcoxon test *p*-value and Benjamini–Hochberg FDR correction *q*-value; (**B**) Association of *SETD7* differential expression (high vs. low) with tissue-infiltrating immune and stromal cell populations using mcp_count method [48] from immunodeconv package and TCGA-BRCA data overall.

**Figure 3 cancers-14-06029-f003:**
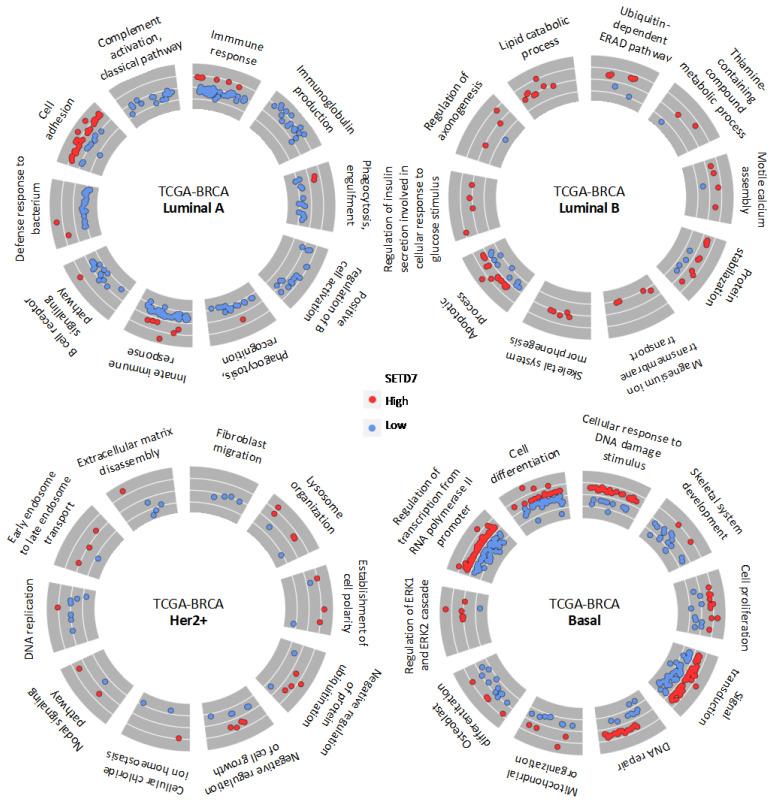
Top 10 biological processes overrepresented among the genes associated with *SETD7* differential expression (high and low) and unique for each subtype.

**Figure 4 cancers-14-06029-f004:**
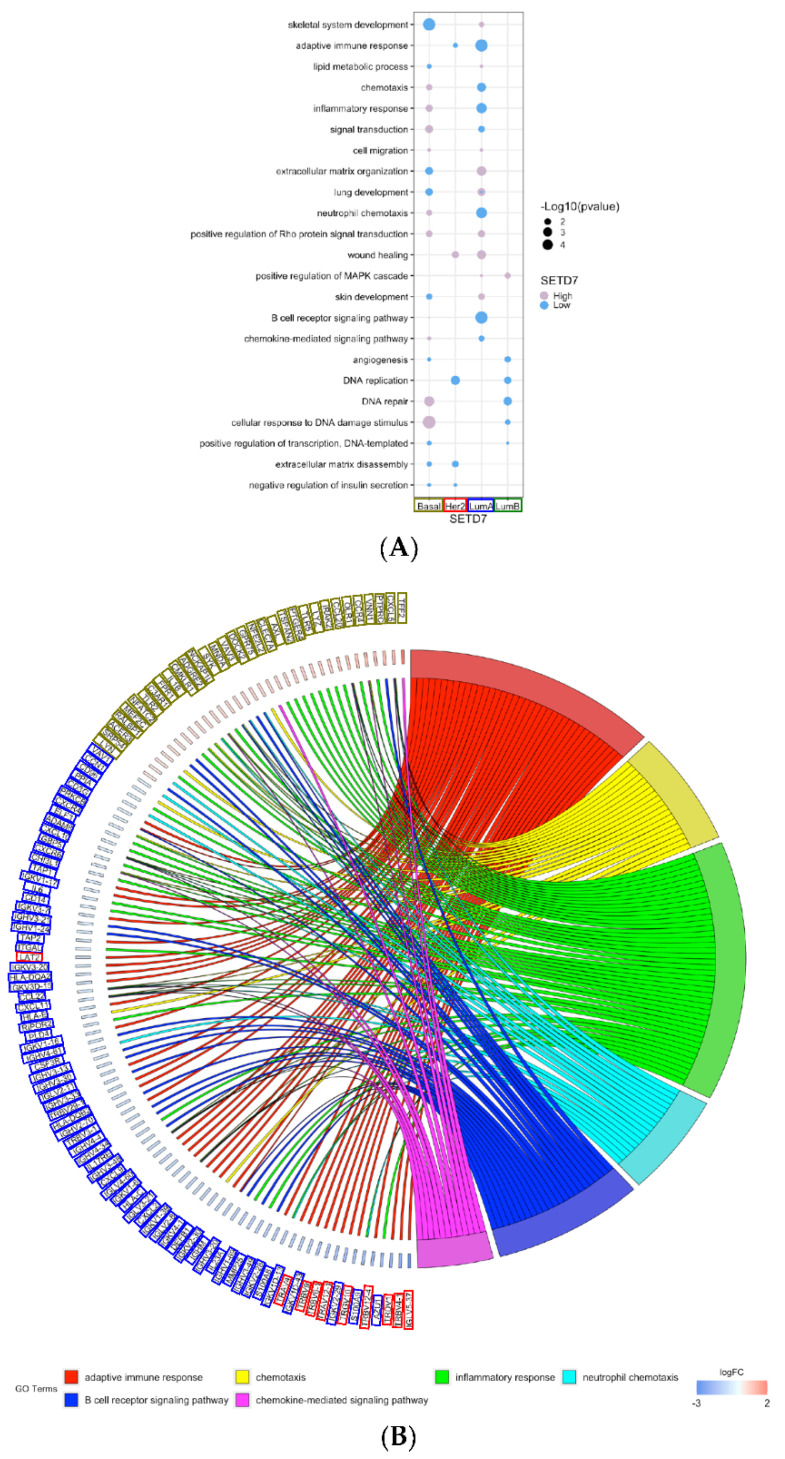
Analysis of the shared biological processes overrepresented among the differentially expressed genes associated with SETD7 (high and low) and unique for each subtype. (**A**) Bubble plot representing all shared biological processes from the unique genes for each subtype; (**B**) GoChord showing all unique genes representing shared immune-related biological processes.

**Figure 5 cancers-14-06029-f005:**
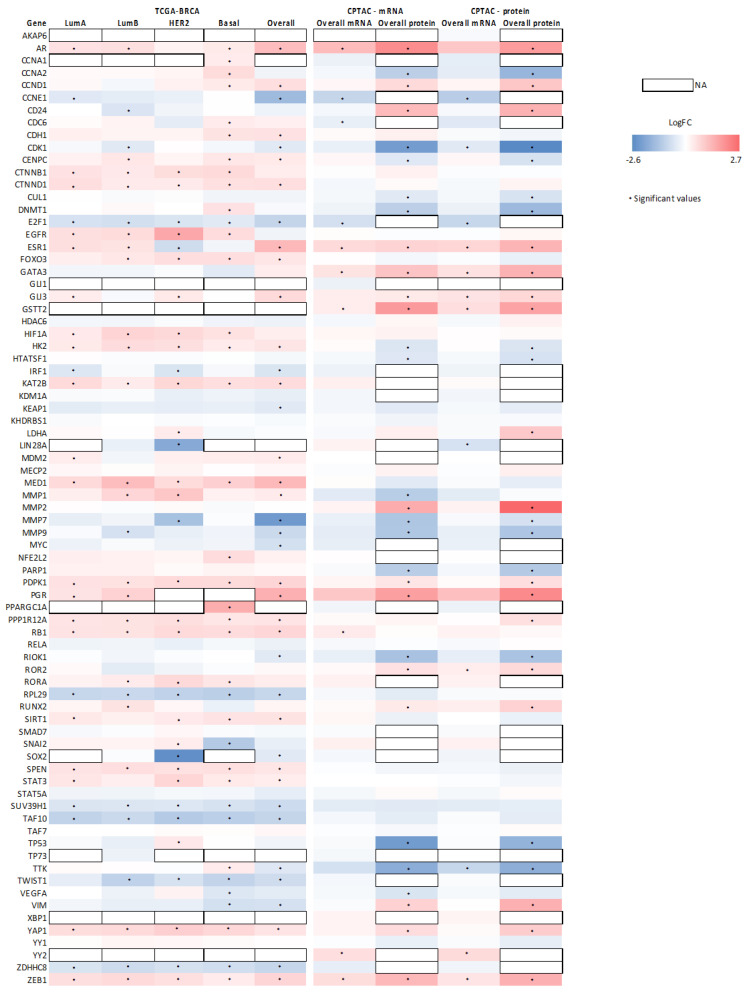
Heatmap showing the association of several genes of interest, including known SETD7 targets, with *SETD7* differential expression using the TCGA-BRCA dataset (analysis performed both by subtype and by pooling all BC samples—overall) and CPTAC dataset (SETD7 stratified by mRNA or protein; analysis performed by pooling all BC—overall). Genes enriched in low-SETD7 group have negative log2 fold change (blue) and the ones enriched in the high-SETD7 group have positive log2 fold change (red). Genes with a significant adjusted *p*-value (Bonferroni post hoc, <0.05) have a star. Genes that were not present for a particular condition are represented as not available (NA, white boxes). Some genes were not detected in any dataset and were excluded from the figure: *ATOH1*, *ESR2* (ERβ), *GATA1*, *NANOG*, *NR1H4* (FXR), and *PDX1*.

**Figure 6 cancers-14-06029-f006:**
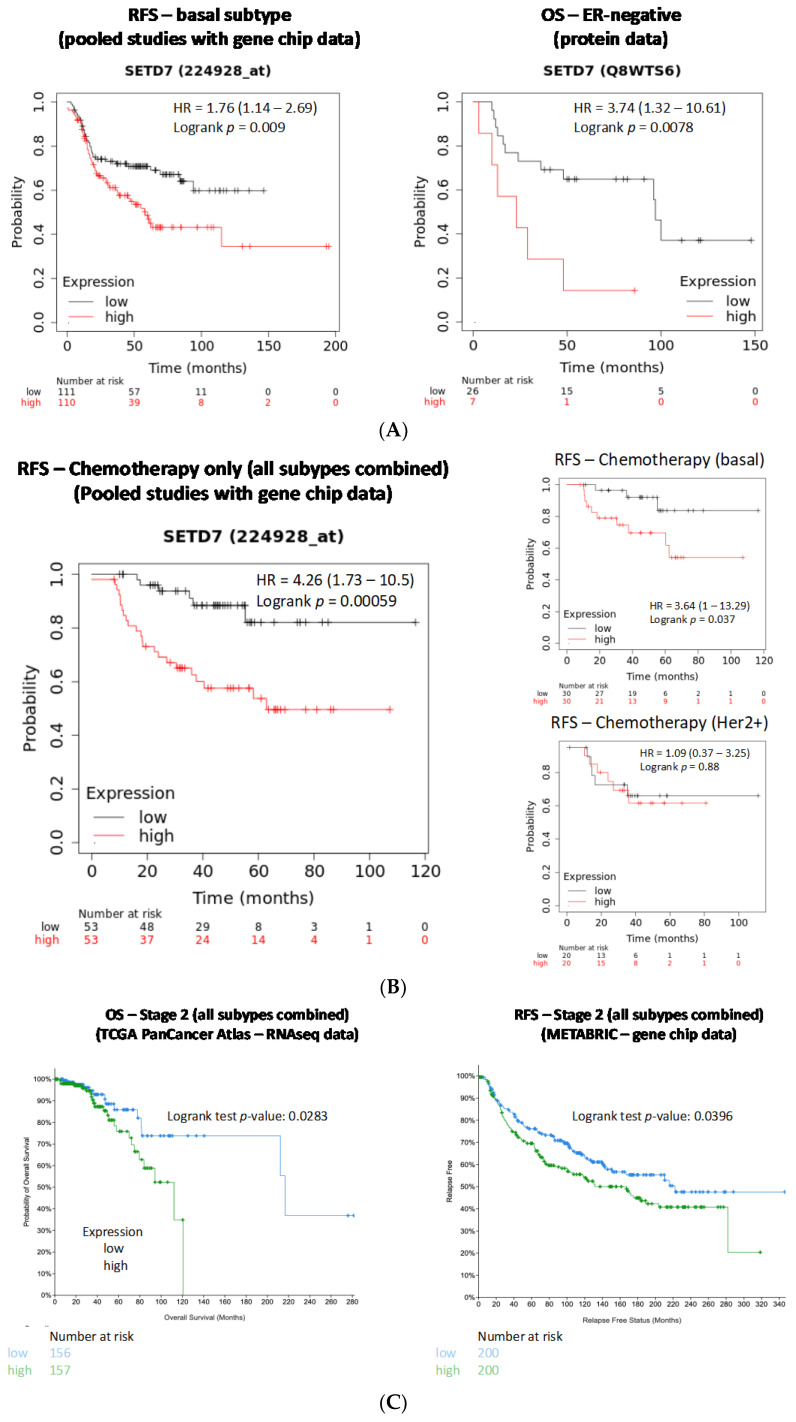
Analysis of the effect of differential SETD7 expression on survival outcomes of BC patients. (**A**) Influence of SETD7 mRNA or protein expression levels on survival outcomes for patients with the basal-like molecular subtype (left panel) or the ER-negative histological subtype (right panel), respectively; (**B**) Influence of *SETD7* expression levels (mRNA) for RFS outcomes following chemotherapy, combining all subtypes (left panel) or by subtype (right panel); (**C**) Influence of *SETD7* expression levels (mRNA) on survival outcomes for stage 2 tumours, using TCGA PanCancer Atlas (RNA-seq, left panel) or METABRIC cohort (gene chip, right panel). OS—overall survival; RFS—relapse/recurrence-free survival.

**Figure 7 cancers-14-06029-f007:**
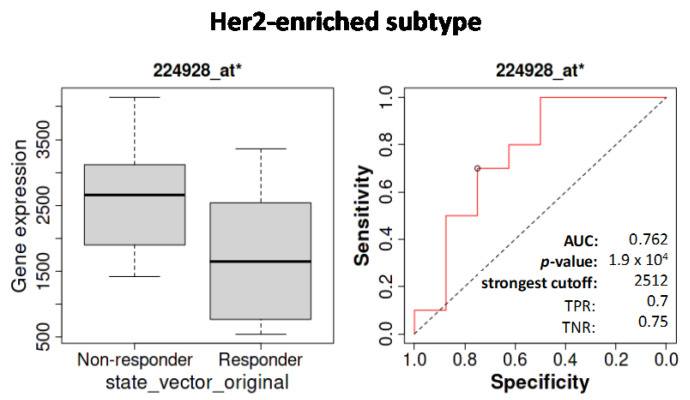
ROCplotter analysis to study the correlation between *SETD7* mRNA expression (microarray data) and 5-year RFS for patients receiving any anti-Her2 therapy (29 responders and 21 non-responders). AUC—area under the curve; TNR—true-negative rate; TPR—true-positive rate.

**Table 1 cancers-14-06029-t001:** *SETD7* expression per subtype. Significant values are highlighted in bold. Chi-squared test *p*-value and Benjamini–Hochberg FDR correction *q*-value. NA—not available; nSETD7 DE—number of samples with differentially expressed SETD7; nTotal—total number of samples.

cBioPortal (nSETD7 DE/n Total Samples)	PAM50	Luminal A	Luminal B	Her2-Enriched	Basal	Normal-Like
CPTAC-RNA (61/122)	***p* = 1.59^−5^** ***q* = 9.85^−5^**	High	Low	High	Low	High
CPTAC-protein (61/122)	***p* = 2.04^−6^** ***q* = 1.58^−5^**	High	Low	High	Low	Unchanged
METABRIC (952/2976)	***p* = 4.23^−3^** ***q* = 0.07**	High	Unchanged	High	Unchanged	Unchanged
SMC (84/187)	***p* = 1.45^−7^** ***q* = 2.33^−6^**	High	High	High	Low	Unchanged
TCGA PanCancer Atlas (541/1084)	***p* < 10^−10^** ***q* < 10^−10^**	High	High	High	Low	Unchanged

**Table 2 cancers-14-06029-t002:** SETD7 association with stromal, immune, stemness and hypoxia scores. Significant values are highlighted in bold: Wilcoxon test *p*-value and Benjamini–Hochberg FDR correction *q*-value. NA—not available.

Scores	CPTAC			TCGA PanCancer Atlas			High SETD7 Correlates with
	RNA	Protein		Overall	Luminal A	Luminal B	
	Overall		Luminal A				
xCell Strommal	***p* = 4.06^−4^** ***q* = 2.17^−3^**	***p* = 1.26^−4^** ***q* = 8.05^−4^**	***p* = 2.09^−3^** ***q* = 0.01**	NA	NA	NA	High
ESTIMATE Strommal	***p* = 1.83^−3^** ***q* = 7.33^−3^**	***p* = 5.96^−4^** ***q* = 2.38 ^−3^**	***p* = 2.25^−3^** ***q* = 0.01**	NA	NA	NA	High
xCell Immune	***p* = 0.01** ***q* = 0.03**	***p* = 8.83^−3^** ***q* = 0.03**	***p* = 0.68** ***q* = 0.81**	NA	NA	NA	Low
Stemness	***p* = 0.01** ***q* = 0.03**	***p* = 2.33^−4^** ***q* = 5.96^−4^**	***p* = 0.06** ***q* = 0.20**	NA	NA	NA	Low
Buffa Hypoxia	NA	NA	NA	***p* < 1.00^−10^** ***q* < 1.00^−10^**	***p* < 1.00^−10^** ***q* < 1.00^−10^**	***p* = 5.82^−4^** ***q* = 0.01**	Low
Winter Hypoxia	NA	NA	NA	***p* < 1.00^−10^** ***q* < 1.00^−10^**	***p* = 1.11^−8^** ***q* = 2.51^−7^**	***p* = 6.36^−4^** ***q* = 0.01**	Low

## Data Availability

Not applicable.

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
