# Peer review of "SETD7 Expression Is Associated with Breast Cancer Survival Outcomes for Specific Molecular Subtypes: A Systematic Analysis of Publicly Available Datasets"

_cancers, 2022, doi:10.3390/cancers14246029_

Round 1
Reviewer 1 Report (Previous Reviewer 1)
The manuscript has been consistently improved and questions duly answered.
This manuscript is a resubmission of an earlier submission. The following is a list of the peer review reports and author responses from that submission.
Round 1
Reviewer 1 Report
The study presented is well designed and aims to elucidate the intricate biology associated with SETD7. Despite obtaining very robust results and a well-designed methodology, there are some issues to pay attention to, especially concerning clinical relevance. As questions, analysis of differential expression of SETD7 by stages (I, II, and III) was not performed, as was done for molecular subtypes.
One of the main findings concerns the response to anti-HER2. However, it was not described how the authors selected patients who received this therapy.
Although there are critical statistical differences, the strength of these differences remains in doubt. For example, it is visually possible to observe more significant differences in SETD7 expression between healthy and tumor tissue (Figure 1A), but not so much by subtype (Figure 1C), which raises doubts as to whether this may have clinical relevance. Please add effect sizes such as omega squared for testing averages, Cramer V for association testing, and so on.
In line with the previous reasoning, at no time were the study's limitations discussed. For example, the analyses of survival and prognosis are all univariable, and multivariable adjustment is unavailable by the tools used. This implies that it is not possible to say concretely whether the level of expression of SETD7 is an independent factor of diagnosis or resistance to therapies. This is more critical about the statements regarding resistance since the measures used were indirect (survival), unlike parameters such as RECIST. Even though we performed a series of analyzes stratified by subgroups, there is still no adjustment for several factors, including whether the therapies were completed correctly.
Other minor revisions:
In the analysis referring to survival and prognosis, replace the adjectives "good" and "bad" with the respective Hazard Ratios (HR) with their confidence intervals, as well as reciprocated by the KM Plotter.
Check image quality. The text of some supplementary figures cannot be understood, and zooming in displays blurry text.
In line 84, the spelling of overall survival is inverted (survival overall).
As a rule, the results are not presented in the final paragraph of the Introduction, much less the p-values. I suggest a restructuring to focus on the objectives of the present manuscript. Similarly, avoid statistics in the Discussion.
Explain abbreviations at first appearances, such as FPM, TPM, and RSEM (line 113)
DFS acronym was not defined (line 124).
SETD7 is misspelled as SET7 in line 391.
Author Response
We appreciate the time dedicated to review our work. We have addressed all the constructive critics to the best of our capabilities and provide a point-by-point response to each comment. Please note that the responses included in the revised version are colour coded [blue for reviewer 1 (this reviewer), green for reviewer 2 and orange for reviewer 3].
Main concerns
1. “The study presented is well designed and aims to elucidate the intricate biology associated with SETD7. Despite obtaining very robust results and a well-designed methodology, there are some issues to pay attention to, especially concerning clinical relevance. As questions, analysis of differential expression of SETD7 by stages (I, II, and III) was not performed, as was done for molecular subtypes.”
Response: We understand the concern and would like to clarify why the analysis by stages was not done as for molecular subtypes:
i. We compared tumours with high vs low SETD7 differential expression and tumour stage but found no difference related with stage. This is now further highlighted in lines 194-196 and Table S2.
ii. We did observe a correlation between high SETD7 and bad prognosis for stage 2 tumours specifically. While this would support analysis by stage as suggested by the reviewer, the majority (>60%) of these patients (TCGA dataset) had stage II tumours along with the fact that this group was mostly comprised by luminal tumour. Thus, the analysis by stage would not improve our information with regards to the analysis already carried out in the luminal molecular subtype. To clarify this reasoning, this is now highlighted in lines 458-463 of the manuscript.
2. One of the main findings concerns the response to anti-HER2. However, it was not described how the authors selected patients who received this therapy.
Response: We agree that this was not described in sufficient detail. The analysis was carried out using the ROC plotter online tool (https://www.rocplot.org/), and the selection of patients receiving therapy was based on the information associated with each sample in the database. For instance, the response to therapy was determined using either author-reported pathological complete response data (n = 1775: responders = 639; non-responders = 1136) or relapse-free survival status at 5 years (n = 1329: responders = 978; non-responders = 351). The method section has now been expanded to reflect this information (lines 111-118). For the specific case illustrated in Figure 7, there were 29 responders and 21 non-responders, and they were determined based on the relapse-free survival at 5 years, as indicated in the figure legend (lines 489-491).
3. Although there are critical statistical differences, the strength of these differences remains in doubt. For example, it is visually possible to observe more significant differences in SETD7 expression between healthy and tumor tissue (Figure 1A), but not so much by subtype (Figure 1C), which raises doubts as to whether this may have clinical relevance. Please add effect sizes such as omega squared for testing averages, Cramer V for association testing, and so on.
Response: We understand the concern and have done our best to follow the Reviewer’s suggestion. For the data in Figure 1B, we calculated the mean differences of each comparison (Cohen´s d) and the confidence intervals associated with them. This is now included as new Figure S3. We further detail the effect sizes (see lines 183-192). Importantly, the effect sizes between basal vs luminal A or B, and Her2-enriched vs basal were large, supportive of relevant clinical differences. Figures 1A and 1C, however, were generated using online tools (TNMplot and cBioportal, respectively) that only allow the in-built statistics (included in our initial submission) and we were unable to perform further analyses.
4. In line with the previous reasoning, at no time were the study's limitations discussed. For example, the analyses of survival and prognosis are all univariable, and multivariable adjustment is unavailable by the tools used. This implies that it is not possible to say concretely whether the level of expression of SETD7 is an independent factor of diagnosis or resistance to therapies. This is more critical about the statements regarding resistance since the measures used were indirect (survival), unlike parameters such as RECIST. Even though we performed a series of analyzes stratified by subgroups, there is still no adjustment for several factors, including whether the therapies were completed correctly.
Response: The Reviewer is correct that we cannot adjust for all factors using these tools. To reflect this limitation, we have added a section in the discussion (please see new lines 591 – 596). The same approach that we use has consistently been used in the SETD7 literature arriving to conclusions regarding SETD7 diagnosis and prognosis potential using one or few datasets. Our unique variable is SETD7 differential expression, and the impact of clinical factors is shown in tables 1 and S2 where we analyse the correlation of SETD7 with the available clinical attributes. We agree that correction for confounding effects is desirable to propose SETD7 as an independent prognostic marker, however the datasets did not contain enough data to do this kind of analysis. As an example, the enrolment form for each of the patients analysed in TCGA (which is the most complete dataset), show only few clinical attributes, such as menopausal status, age of diagnosis, ethnicity, tumour stage and grade. The correlation between all these clinical attributes and SETD7 differential expression were analysed in cBioPortal and the results are shown in Table S2. The follow-up form is also attached for the reviewers’ evaluation. Concerning resistance, the most widely parameter used by most clinicians is to report progression or relapse free survival. We agree that this is not necessarily the same as resistance and have modified the statements accordingly (lines 563, 575 and 585).
Other comments from Reviewer 1:
- In the analysis referring to survival and prognosis, replace the adjectives "good" and "bad" with the respective Hazard Ratios (HR) with their confidence intervals, as well as reciprocated by the KM Plotter.
Response: The Hazard Ratios were added as suggested.
- Check image quality. The text of some supplementary figures cannot be understood, and zooming in displays blurry text.
Response: This has been adjusted.
- In line 84, the spelling of overall survival is inverted (survival overall).
Response: This line was now removed from the introduction (as requested in point 4).
- As a rule, the results are not presented in the final paragraph of the Introduction, much less the p-values. I suggest a restructuring to focus on the objectives of the present manuscript. Similarly, avoid statistics in the Discussion.
Response: Results have been deleted from the Introduction and statistics removed from the Discussion.
- Explain abbreviations at first appearances, such as FPM, TPM, and RSEM (line 113)
Response: Done (please see new lines 103-105)
- DFS acronym was not defined (line 124).
Response: This acronym was defined in line 60 of the initial submission. It has now been highlighted.
- SETD7 is misspelled as SET7 in line 391.
Response: Fixed

Reviewer 2 Report
General comments
The study described in the manuscript addresses an interesting aspect related to new biomarkers with prognostic and predictive value in breast cancer. However, but the findings are limited to expand the in silico datamining previously reported in the literature (Huang R et al., 2017), without bringing any further validation in clinical specimens. In silico, data mining is an important and valuable tool to generate new hypotheses; however, the findings should be further validated.
Specific comments
Figure 1 – The number of patients (N) should be provided in all box-plots.
In the discussion section, the authors should highlight the limitations of their study.
The English have to be reviewed to improve the quality and clarity of the manuscript. To improve the English language, the authors should consider having their manuscript reviewed by someone fluent in English or professional help for revising the document.
Author Response
We appreciate the time dedicated to review our work. We have addressed all the constructive critics to the best of our capabilities and provide a point-by-point response to each comment. Please note that the responses included in the revised version are colour coded [blue for reviewer 1, green for reviewer 2 (this reviewer) and orange for reviewer 3].
Main concerns
- “The study described in the manuscript addresses an interesting aspect related to new biomarkers with prognostic and predictive value in breast cancer. However, but the findings are limited to expand the in silicodatamining previously reported in the literature (Huang R et al., 2017), without bringing any further validation in clinical specimens. In silico, data mining is an important and valuable tool to generate new hypotheses; however, the findings should be further validated.”
Response: We agree that validations of the prognostic and predictive value of SETD7 is important. However, the focus of our work is to bring clarification to the inconsistent findings from previous in silico studies. Most of these studies were unclear regarding which datasets had been used (often limited to one or a few smaller sets) and provided incomplete description of the analysis carried out. This was one of our main findings from a previous systematic review (Monteiro, et al, Cancers 2022 https://www.mdpi.com/2072-6694/14/6/1414). In our current work, we have systematically analysed all the datasets and provided a thorough description of the datasets and methods used in our analysis, together with the application of a variety of tools, some of which had not been previously used such as the CPTAC dataset and immune deconvolution methods. This alone is of importance to resolve the current inconsistencies in the literature. We present novel findings related to molecular subtype, and identify biological processes associated with SETD7 differential expression not previously investigated. We agree that further validation in clinical specimens with be needed to confirm these results. Still, these exceed the aim of this work as stated in the title of the manuscript and would also require a substantial amount of time. We have added a sentence raising this limitation and the need for future studies in the Discussion (lines 531-532).
Specific comments
- Figure 1 – The number of patients (N) should be provided in all box-plots.
Response: Done
- In the discussion section, the authors should highlight the limitations of their study.
Response: This has been included (please see new lines 530 - 531) and also the response to the reviewer 1 in lines 591-596
- The English have to be reviewed to improve the quality and clarity of the manuscript. To improve the English language, the authors should consider having their manuscript reviewed by someone fluent in English or professional help for revising the document.
Response: We have carefully reviewed the entire manuscript to improve the English language

Reviewer 3 Report
Dear authors,
thank you for the very interesting manuscript.
MAJOR REVISIONS:
- The methods are adequately described but the results, as well as the conclusions, are not clearly presented. For this purpose, it could be useful and interesting to show specifically how the impact of SETD7 should be evaluated in the context of different molecular subtypes and how its expression is related to prognosis. In this regard, the conclusions are not adequately supported by the results, in particular you might add the correlation data between the expression of SETD7, Breast Cancer (BC) subtypes, and BC Survival Outcomes to underline its potential prognostic value. In particular, the data related to long-term outcomes should be better explained and supported.
- Line 503: I would compare data on the role of RB1 and pRb in osteosarcoma and breast cancer, respectively, and relevant references should be inserted, or unclear parallelism should be removed.
- Line 511: I would add data on the hypoxia-related role of SETD7, based on the response to chemotherapy, of different breast cancer subtypes. Related references, on the predictive role of SETD7 expression, should be included by the authors.
MINOR REVISIONS:
- It would be interesting if the authors could expand, in the discussion, the concept of post-transcriptional regulation of SETD7.
- In the discussion, the long digression on immunotherapy and resistance to endocrine therapy, according to SETD7, is not clear also considering the undefined role of immune checkpoint inhibitors (ICIs) in luminal-like breast cancer.
Author Response
REVIEWER 3
We appreciate the time dedicated to review our work. We have addressed all the constructive critics to the best of our capabilities and provide a point-by-point response to each comment. Please note that the responses included in the revised version are colour coded [blue for reviewer 1, green for reviewer 2 and orange for reviewer 3 (this reviewer)].
Main concerns
- “The methods are adequately described but the results, as well as the conclusions, are not clearly presented. For this purpose, it could be useful and interesting to show specifically how the impact of SETD7 should be evaluated in the context of different molecular subtypes and how its expression is related to prognosis. In this regard, the conclusions are not adequately supported by the results, in particular you might add the correlation data between the expression of SETD7, Breast Cancer (BC) subtypes, and BC Survival Outcomes to underline its potential prognostic value. In particular, the data related to long-term outcomes should be better explained and supported.”
Response: The Discussion has now been restructured to better describe the findings per subtype and make the text clearer. In addition, the graphical abstract is intended to facilitate the understanding of results and conclusions.
- Line 503: I would compare data on the role of RB1 and pRb in osteosarcoma and breast cancer, respectively, and relevant references should be inserted, or unclear parallelism should be removed.
Response: We agree, this parallelism between two biologically different tumors has been removed.
- Line 511: I would add data on the hypoxia-related role of SETD7, based on the response to chemotherapy, of different breast cancer subtypes. Related references, on the predictive role of SETD7 expression, should be included by the authors.
Response: We realize the sentence in question was misleading and have now removed the reference to Hif1a and kept only the relationship between DNA damage response and chemotherapy.
Other comments from Reviewer 3:
- It would be interesting if the authors could expand, in the discussion, the concept of post-transcriptional regulation of SETD7.
Response: We rearranged the sentence in question.
- In the discussion, the long digression on immunotherapy and resistance to endocrine therapy, according to SETD7, is not clear also considering the undefined role of immune checkpoint inhibitors (ICIs) in luminal-like breast cancer.
Response: We agree and have now clarified this idea (line 540-547).
